

# CD80 and CTLA-4 as diagnostic and prognostic markers in adult-onset minimal change disease: a retrospective study

Bing Zhao[1,*], Hui Han[2,*], Junhui Zhen[3], Xiaowei Yang[1], Jin Shang[4], Liang Xu[1] and Rong Wang[1]

[1] Department of Nephrology, Shandong Provincial Hospital Affiliated to Shandong University, Jinan, China
[2] Department of Intensive Care Unit, Shandong University Qilu Hospital, Jinan, China
[3] School of Medicine, Shandong University, Jinan, China
[4] Department of Nephrology, The First Affiliated Hospital of Zhengzhou University, Zhengzhou, China
* These authors contributed equally to this work.

## ABSTRACT

**Background:** Minimal change disease (MCD) is a form of idiopathic nephrotic syndrome. Compared to children, adult-onset MCD patients are reported to have delayed responses to glucocorticoid treatment. Several studies of children have suggested detecting urinary CD80 levels to diagnose MCD. There are no effective diagnostic methods to distinguish steroid-sensitive MCD from steroid-resistant MCD unless treatments are used.

**Methods:** In a total of 55 patients with biopsy-proven MCD and 26 patients with biopsy-proven idiopathic membranous nephropathy, CD80 and cytotoxic T-lymphocyte antigen-4 (CTLA-4) levels in serum, urine and renal tissue were analyzed.

**Results:** Steroid-sensitive MCD patients in remission had lower urinary CD80 levels and higher CTLA-4 levels than patients in relapse (156.65 ± 24.62 vs 1066.40 ± 176.76 ng/g creatinine; $p < 0.0001$), (728.73 ± 89.93 vs 151.70 ± 27.01 ng/g creatinine; $p < 0.0001$). For MCD patients in relapse, mean urinary CD80 level was higher, and CTLA-4 level was lower for those who were steroid-sensitive than those who were steroid-resistant (1066.40 ± 176.76 vs. 203.78 ± 30.65 ng/g creatinine; $p < 0.0001$), but the mean urinary CTLA-4 level was lower (151.70 ± 27.01 vs. 457.83 ± 99.45 ng/g creatinine; $p < 0.0001$). CD80 expression in glomeruli was a sensitive marker to diagnose MCD. The absent or minimal expression of CTLA-4 in glomeruli could distinguish steroid-sensitive MCD from steroid-resistant MCD.

**Conclusions:** Glucocorticoid treatment may result in complete remission for only MCD patients with strongly positive CD80 expression and negative CTLA-4 expression in glomeruli, or higher urinary CD80 level and lower CTLA-4 level.

Corresponding author
Rong Wang, sdwangrong@sina.cn

## BACKGROUND

Minimal change disease (MCD) is a common form of idiopathic nephrotic syndrome that accounts for 10–15% of nephrotic diseases in adults (*Zech et al., 1982*; *Waldman et al., 2007*). MCD is a serious and challenging disease for adult-onset patients, of which around 25% are steroid-resistant. Compared to children, adult-onset MCD patients were reported to have higher risks of acute kidney injury and delayed responses to glucocorticoids treatment (*Huang et al., 2001*; *Szeto et al., 2015*). It has been reported that 73% of MCD patients experienced at least one relapse, of which 28% suffered frequent relapses (*Waldman et al., 2007*).

Nephrologists often put forward two options to MCD patients who relapse again: either receive the drug therapy (i.e., full-dose glucocorticoid therapy, cytotoxic drugs), or seek further diagnostic tests, such as renal biopsy. Most patients tend to choose the former. These choices could result in ineffective treatments and/or increased adverse drug reactions (*Barbarino et al., 2013*; *Grimm et al., 2006*). The mechanism of steroid-resistant MCD is unknown. The characteristics of patients (e.g., age or medication compliance) and pathologic misdiagnosis, which may be due to similar imaging features of MCD, focal segmental glomerulosclerosis (FSGS), and stage I of idiopathic membranous nephropathy (IMN) under light microscopy are possible causes. Long-term prognosis may not be favorable, as indicated in a study in which a large number of patients with adult-onset MCD were found to have FSGS on a second kidney biopsy and experienced progression to end-stage renal disease (ESRD) or death (*Szeto et al., 2015*). According to the North American Pediatric Renal Trials and Collaborative Studies, steroid-resistant nephrotic syndrome constitutes the second most frequent cause of ESRD in the first two decades of life (*Smith, Martz & Blydt-Hansen, 2013*). Therefore, early-stage identification of steroid-resistant nephritic syndrome is needed in MCD patients.

The pathogenesis of MCD remains unclear; however, several hypotheses have been proposed. For several decades, MCD has been considered a T-cell disorder, and increased levels of several cytokines were also suggested. Recently, a proposed "two-hit" theory proposed the induction of CD80 (B7-1) and regulatory T-cell dysfunction (*Shimada et al., 2011*), with or without impaired autoregulatory function of podocytes. Several studies of children suggested detecting urinary CD80 level to distinguish MCD from FSGS (*Cara-Fuentes et al., 2014b*; *Garin et al., 2009*, *2010*; *Ling et al., 2015*), but there have been few studies of adult-onset MCD. Abatacept (cytotoxic T-lymphocyte–associated antigen 4–immunoglobulin fusion protein [CTLA-4–Ig]), a costimulatory inhibitor that targets CD80, has been used in CD80-accociated nephropathy (*Yu et al., 2013*; *Trimarchi, 2015*). However, its effectiveness is controversial (*Yu et al., 2013*; *Norlin et al., 2016*; *Garin et al., 2015*). Here, we aimed to evaluate whether CD80 and CTLA-4 could be diagnostic and prognostic markers in adult-onset MCD, and whether these markers could be useful for predicting the effectiveness of single-glucocorticoid treatment in adult-onset MCD patients.

## MATERIALS AND METHODS

### Patients

All patients were followed at Shandong Provincial Hospital affiliated to Shandong University. Patients over 14 years old were recommended by specialist physicians instead of pediatricians in China and received diagnosis and treatment following standards for adults. The study was approved by the Institutional Review Board of Shandong Provincial Hospital affiliated to Shandong University (No. 2014-022). Before participation in this study, written informed consent was obtained from all patients and their parents/guardians.

### Inclusion and exclusion criteria

Our research subjects were first selected from the hospitalized patients in the nephrology department of Shandong Provincial Hospital affiliated to Shandong University between March and November 2014. Our inclusion criteria were: (1) diagnosed as nephrotic syndrome patients by specialist physicians; (2) age no less than 14 years old; (3) renal pathological diagnoses as MCD or IMN; (4) estimated glomerular filtration rate (eGFR) calculated by Creatinine Equation (*Levey et al., 2009*) higher than 60 ml/min per 1.73 $m^2$. The exclusion criteria were: (1) pregnant women, tumor patients, and urinary system lithiasis patients; (2) lost to follow-up without any prognosis. The IMN patients were the control group.

### Definitions

**Complete remission (CR):** Urinary protein excretion < 0.3 g/d or urine protein:creatinine ratio (uPCR) < 30 mg/mmol. **Partial Remission:** Urinary protein excretion < 3.5 g/d or uPCR < 350 mg/mmol and a 50% or greater reduction from peak values, accompanied by an improvement or normalization of the serum albumin concentration. **Relapse:** Proteinuria > 3.5 g/d or uPCR > 350 mg/mmol. **Steroid-resistance:** Failure to achieve remission after 8 weeks of corticosteroid therapy. **Steroid-sensitivity**: Achieved CR during 8 weeks of corticosteroid therapy. Renal pathologic diagnoses of patients were established using light and electron microscopy by two pathologists. **Glucocorticoid treatment:** Prednisone one mg/kg per day for 8 weeks, and then reduce the 10% of total dosage every 2 weeks (*Kidney Disease: Improving Global Outcomes (KDIGO) Glomerulonephritis Work Group, 2012*; *Meyers & Kaplan, 2001*; *Wang, 2009*).

### CD80 and CTLA-4 measurements

Serum and urinary CD80 and CTLA-4 were detected when patients were in relapse or complete/partial remission, and 24 h urinary protein, uPCR and serum albumin were measured in the same day. CD80 and CTLA-4 levels were measured using a commercially available ELISA kit (Bender MedSystems, eBioscience, Vienna, Austria), and results were adjusted for urinary creatinine excretion. Urinary creatinine level and protein and serum albumin levels were measured using an autoanalyzer.

### Immunohistochemistry

Kidney samples were obtained from excess tissue corresponding to kidney nephrectomy specimens donated to the biobank of Shandong University after diagnostic evaluation.

**Table 1 Laboratory data of steroid-sensitive MCD patients in remission and relapse.**

| Laboratory data | Relapse | Remission |
|---|---|---|
| Serum albumin (g/l) | 20.04 ± 1.19[a] | 27.83 ± 1.19 |
| 24 h urinary protein (g) | 5.33 ± 0.35 | 0.34 ± 0.09 |
| Urinary CD80 (ng/g creat) | 1,066.40 ± 176.76 | 156.65 ± 24.62 |
| (95% CI[b]) | (705.90–1426.91) | (106.44–206.86) |
| Urinary CTLA-4 (ng/g creat) | 151.70 ± 27.01 | 728.73 ± 89.93 |
| (95% CI) | (96.61–206.79) | (545.33–912.13) |
| Serum CD80 (ng/l) | 0.87 ± 0.12 | 0.63 ± 0.08 |
| (95% CI) | (0.63–1.11) | (0.47–0.79) |
| Serum CTLA-4 (ng/l) | 0.33 ± 0.07 | 0.33 ± 0.07 |
| (95% CI) | (0.19–0.47) | (0.19–0.46) |
| Glucocorticoids treatment | | |
| None | 19 | 0 |
| 1–7 days | 9 | 11 |
| 8–28 days | 1 | 16 |
| 29–112 days | 0 | 2 |
| >112 days | 3 | 3 |

**Notes:**
[a] Data are mean ± s.e.m.
[b] CI, confidence interval for mean.

Immunohistochemistry involved 5-μm-thick paraffin-embedded tissue sections. The primary antibodies were mouse anti-human monoclonal CD80 (1:150) and CTLA-4 (1:100, both Santa Cruz Biotechnology), and the secondary antibodies were a rabbit anti-mouse Biotin-Streptavidin HRP Detection Systems (Zhongshanjinqiao Biotechnology company, Beijing, China). Sections were counterstained with Carazzi's hematoxylin.

## Statistical analysis

Statistical analyses were performed with SPSS 17.0, and receiver operating characteristic (ROC) curve analyses were performed with Medcalc 17.0. Data were analyzed by $t$-test. Results were considered significant at $p < 0.05$.

## RESULTS

We detected serum and urinary CD80 and CTLA-4 levels by ELISA in 55 patients with biopsy-proven MCD and 26 patients with biopsy-proven IMN.

### Comparison of urinary CD80 and CTLA-4 excretion of steroid-sensitive MCD patients in relapse and remission

We detected all the laboratory data (levels of serum albumin, 24 h urinary protein, serum CD80 and CTLA-4, urinary CD80 and CTLA-4) for all our steroid-sensitive MCD patients when they were in relapse and remission. The laboratory data and glucocorticoid treatments are shown in Table 1.

Urinary CD80 excretion was lower for MCD patients in remission than relapse (156.65 ± 24.62 vs 1066.40 ± 176.76 ng/g creatinine; $p < 0.0001$). However, mean urinary

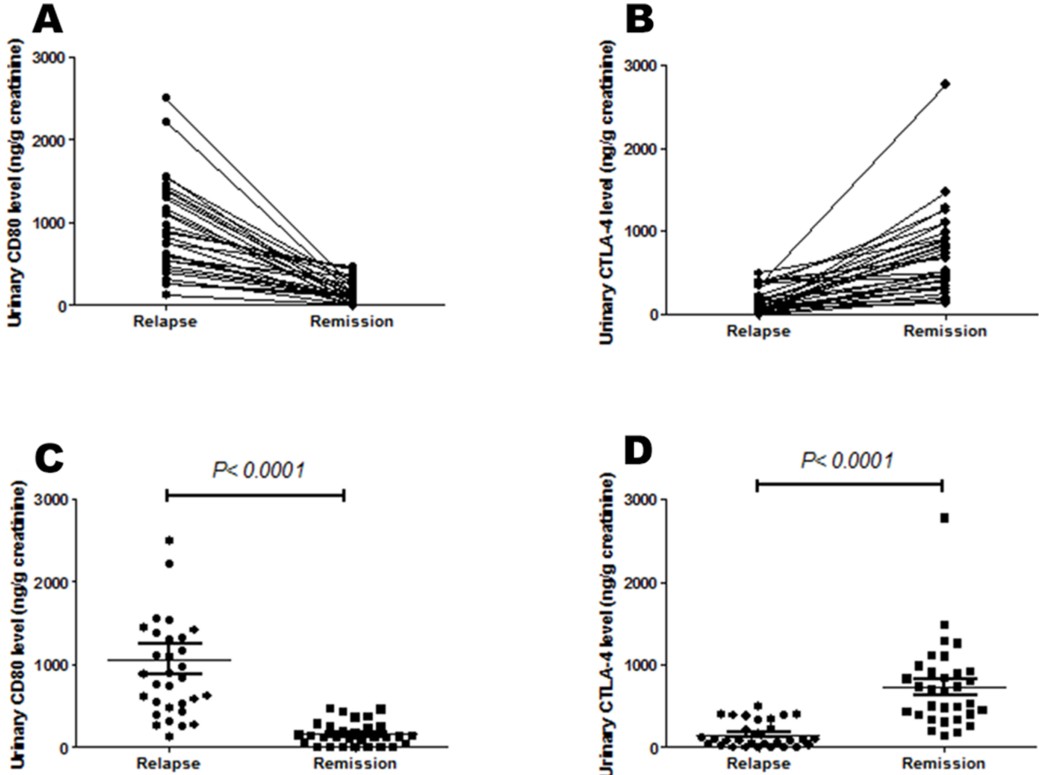

**Figure 1 Urinary CD80 and urinary CTLA-4 levels of steroid-sensitive MCD patients in relapse and remission.** (A) Urinary CD80 excretion was lower for MCD patients in remission than relapse respectively, but (B) urinary CTLA-4 excretion was higher. (C) Mean urinary CD80 excretion was lower for MCD patients in remission than relapse significantly, but (D) mean urinary CTLA-4 excretion was higher.

CTLA-4 levels were greater for MCD patients in remission than relapse (728.73 ± 89.93 vs 151.70 ± 27.01 ng/g creatinine; $p < 0.0001$) (Table 1; Fig. 1).

The area under the ROC curve (AUC) comparing MCD patients in relapse versus remission was 0.957 for urinary CD80 and 0.0.928 for urinary CTLA-4 (Fig. 2), with no significant difference between these two AUCs.

The serum CD80 and CTLA-4 levels of steroid-sensitive MCD patients in relapse were not statistically different from these patients when they were in remission (Table 1).

## Comparison of urinary CD80 and CTLA-4 excretion of steroid-sensitive MCD patients in relapse, steroid-resistant MCD patients in relapse and IMN in relapse

We compared the CD80 and CTLA-4 levels of 32 steroid-sensitive MCD patients in relapse, 23 steroid-resistant MCD patients in relapse, and 26 IMN patients in relapse. The laboratory data and glucocorticoid treatments are shown in Table 2.

The urinary CD80 level of steroid-sensitive MCD patients in relapse was significantly higher than that of steroid-sensitive MCD patients in relapse (1066.40 ± 176.76 vs 203.78 ± 30.65 ng/g creatinine; $p < 0.0001$), and it was also significantly

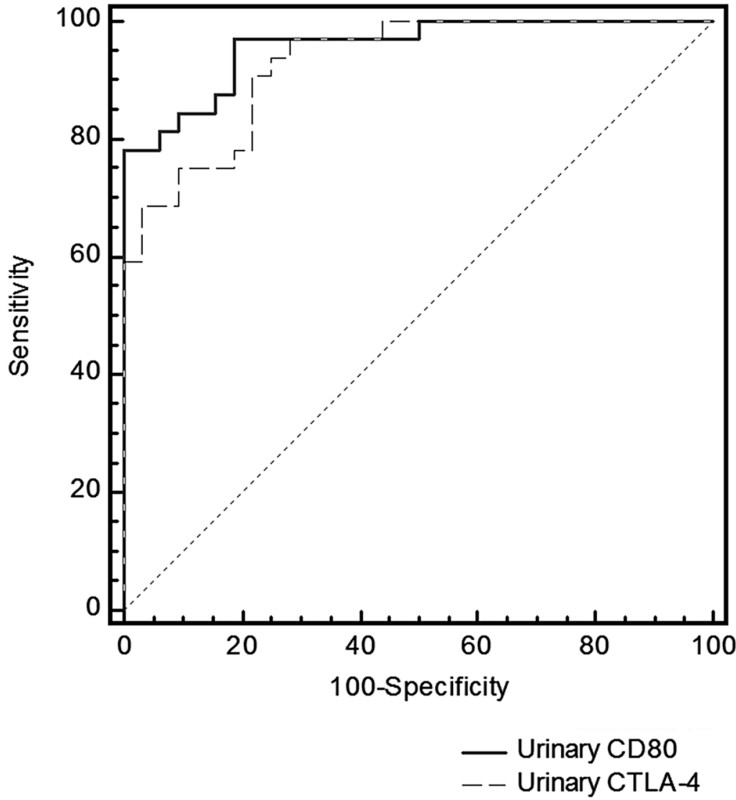

| | AUC | SE | 95%CI |
|---|---|---|---|
| Urinary CD80 | 0.957 | 0.0265 | 0.875 to 0.992 |
| Urinary CTLA-4 | 0.928 | 0.0343 | 0.835 to 0.977 |

**Figure 2 Receiver operating characteristic curves for differentiating relapse and remission in patients with steroid-sensitive MCD.**

higher than that of IMN patients in relapse (1066.40 ± 176.76 vs 294.95 ± 34.08 ng/g creatinine; $p < 0.0001$). Urinary CTLA-4 levels of steroid-sensitive MCD patients in relapse were significantly lower than those of steroid-sensitive MCD patients in relapse (151.70 ± 27.01 vs 457.83 ± 99.45 ng/g creatinine; $p = 0.006$), and they were also significantly lower than those of IMN patients in relapse (151.70 ± 27.01 vs 299.53 ± 47.46 ng/g creatinine; $p = 0.006$). The urinary CD80 levels of steroid-resistant MCD patients in relapse was not different statistically from IMN patients in relapse, as were the urinary CTLA-4 levels of steroid-resistant MCD patients in relapse (Fig. 3). Serum CD80 or CTLA-4 levels did not differ among groups. In comparing steroid-sensitive and steroid-resistant MCD patients in relapse, the AUC for urinary CD80 level was 0.937, and for urinary CTLA-4, it was 0.736 (Fig. 4A). In comparing steroid-sensitive MCD patients and IMN patients in relapse, the AUC for urinary CD80 was 0.867, and for urinary CTLA-4, it was 0.721 (Fig. 4B).
**Table 2 Laboratory data and therapy for steroid-sensitive MCD patients in relapse, steroid-resistant MCD patients in relapse and IMN in relapse.**

| Laboratory data | Steroid-sensitive MCNS patients in relapse | Steroid-resistant MCNS patients in relapse | IMN patients in relapse |
| --- | --- | --- | --- |
| Age (year) (mean ± SD) | 27.63 ± 12.55 | 33.09 ± 17.99 | 38.77 ± 14.63 |
| Serum albumin (g/l) | 20.04 ± 1.19[a] | 21.85 ± 1.70 | 25.91 ± 1.21 |
| 24 h urinary protein (g) | 5.33 ± 2.00 | 4.78 ± 0.38 | 4.45 ± 0.38 |
| Urinary CD80 (ng/g creat) | 1066.40 ± 176.76 | 203.78 ± 30.65 | 294.95 ± 34.08 |
| (95% CI[b]) | (705.90–1426.91) | (140.21–267.34) | (224.77–365.13) |
| Urinary CTLA-4 (ng/g creat) | 151.70 ± 27.01 | 457.83 ± 99.45 | 299.53 ± 47.46 |
| (95% CI) | (96.61–206.79) | (251.58–664.08) | (201.10–397.94) |
| Serum CD80 (ng/l) | 0.87 ± 0.12 | 0.55 ± 0.11 | 0.95 ± 0.22 |
| (95% CI) | (0.63–1.11) | (0.32–0.78) | (0.49–1.46) |
| Serum CTLA-4 (ng/l) | 0.33 ± 0.07 | 0.45 ± 0.12 | 0.31 ± 0.05 |
| (95% CI) | (0.19–0.47) | (0.20–0.71) | (0.20–0.42) |
| Glucocorticoids treatments | | | |
| None | 19 | 3 | 16 |
| 1–7 days | 9 | 0 | 1 |
| 8–28 days | 1 | 0 | 4 |
| 29–112 days | 0 | 6 | 5 |
| >112 days or other immunodepressive therapy | 3 | 14 | 0 |

Notes:
[a] Data are mean ± s.e.m.
[b] CI, confidence interval for mean.

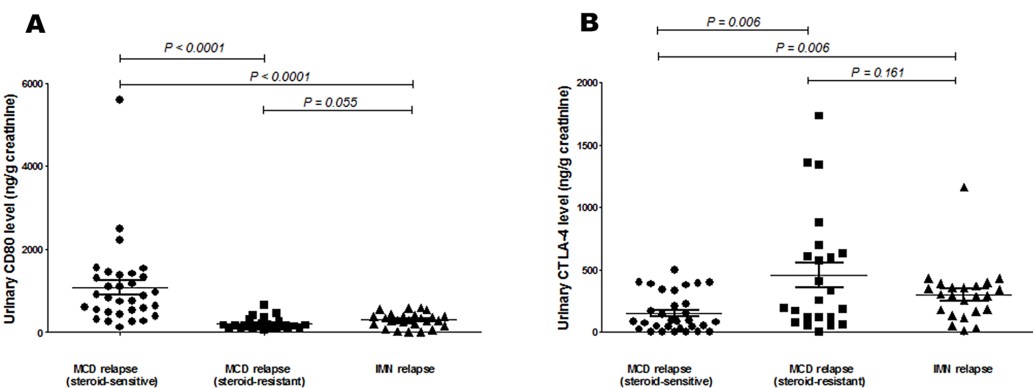

**Figure 3 Comparison of urinary CD80 and CTLA-4 levels of steroid-sensitive MCD patients in relapse, steroid-resistant MCD patients in relapse and IMN in relapse.** (A) Urinary CD80 levels and (B) urinary CTLA-4 levels in steroid-sensitive MCD patients in relapse, steroid-resistant MCD patients in relapse, and IMN patients in relapse.

## CD80 expressed in glomeruli in steroid-sensitive MCD patients in relapse

A limited number of biopsies were available for study, including 17 cases of MCD in relapse, four cases of MCD in remission, and six cases of IMN in relapse. The remission in the four cases of MCD was partial (proteinuria <1 g/24 h) at the time of renal biopsy,

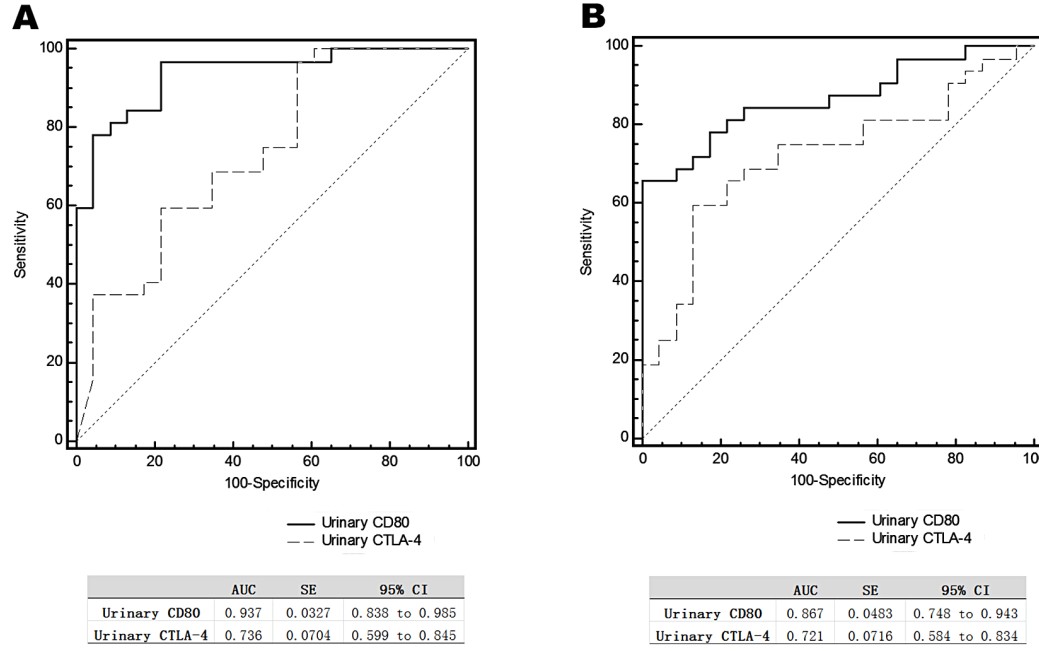

**Figure 4 Receiver operating characteristic curves (ROC) for urinary CD80 with CTLA-4 levels differentiating patients with steroids-sensitive MCD and others.** ROC analysis of urinary CD80 with CTLA-4 levels comparing steroid-sensitive MCD and steroid-resistant MCD patients in relapse (A) and steroid-sensitive MCD and IMN patients in relapse (B).

and several days after biopsy; all four cases showed CR. Glucocorticoids or other immunosuppressive drugs had been used in the IMN cases.

CD80 was present in the glomeruli of patients with steroid-sensitive MCD in relapse, but was minimal or absent for those with steroid-sensitive MCD in remission. CTLA-4 was minimal or absent in the glomeruli of patients with steroid-sensitive MCD in relapse, but was present in the glomeruli of those with steroid-sensitive MCD in remission. Both CD80 and CTLA-4 were present in the glomeruli of patients with steroid-resistant MCD and IMN in relapse, and levels were minimal (Fig. 5).

## DISCUSSION

Few studies have investigated the expression of CD80 and CTLA-4 in adult-onset MCD. Our study demonstrated that urinary CD80 was elevated in MCD during relapse, with levels returning to the low range with disease in remission. Urinary CTLA-4 levels were higher in patients in remission than relapse.

Some studies have investigated these levels in children (*Garin et al., 2009*, *2010*). However, patients with primary adult-onset MCD may have more severe clinical features than pediatric MCD patients. A recent study showed that only 30% of adult Chinese MCD patients achieved CR after initial treatment (*Szeto et al., 2015*). In addition, the long-term prognosis may not be favorable, as indicated in a study finding that a considerable number of patients with adult-onset MCD, showing FSGS on a second kidney biopsy who experienced progression to ESRD or death (*Szeto et al., 2015*). Some studies of patients with adult-onset MCD have reported increased risk of acute kidney injury

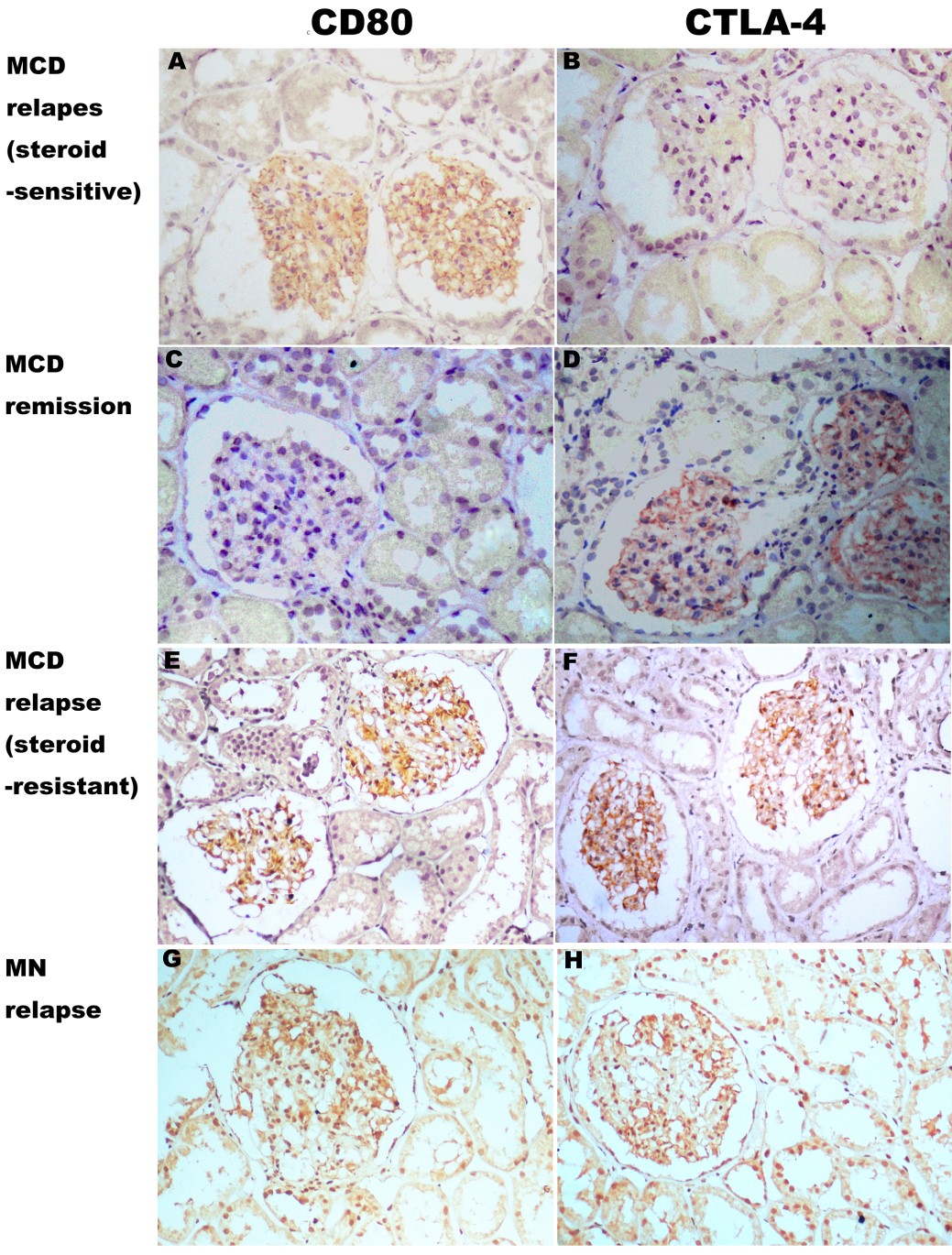

**Figure 5 Expression of CD80 and CTLA-4 in glomerulus of several types of idiopathic nephrotic syndrome.** (A, E and G) CD80 was expressed (brown stain) in the glomerulus from a steroid-sensitive MCD patient in relapse, a steroid-resistant MCD patient in relapse, and an IMN patient in relapse. (C) Minimal stain for CD80 was found in the glomerulus of a steroid-sensitive MCD in partial remission. (D, F and H) CTLA-4 was expressed in the glomerulus from the steroid-resistant MCD patient in remission, the steroid-resistant MCD patient in relapse, and an IMN patient in relapse. (B) CTLA-4 was absent in the glomerulus from the steroid-sensitive MCD patient in relapse. (Immunohistochemistry, original magnification ×200).

(*Huang et al., 2001*; *Szeto et al., 2015*) and delayed response to treatment with glucocorticoid therapy as compared to pediatric MCD patients (*Waldman et al., 2007*; *Korbet, Schwartz & Lewis, 1988*). In our study, the variation in urinary CD80 levels in steroid-sensitive adult-onset MCD was similar to that for pediatric MCD patients. Urinary CD80 and CTLA-4 levels seem to be associated with MCD activity in adults.

We found no differences in serum CD80 or CTLA-4 level in patients in remission than those in relapse; thus, the increased urinary excretion could not be explained by increased serum levels. The immunohistochemical expression of CD80 or CTLA-4 on renal tissue paralleled changes in urinary CD80 or CTLA-4 excretion from relapse to partial remission for steroid-sensitive MCD patients. Urinary CD80 or CTLA-4 may be excreted from the kidney but not blood circulation. *Garin et al. (2010)* tested whether the source of urinary CD80 is podocytes because of differences in molecular weight between soluble CD80 secreted by circulating B cells and whole-cell membrane-associated CD80 (*Greenwald, Freeman & Sharpe, 2005*; *Wong et al., 2005*) and with immunofluorescence studies of renal biopsies.

CD80, also termed B7-1, is a transmembrane protein expressed on the surface of B cells and other antigen-presenting cells. It is one of the major co-stimulators of T-cell activation by binding to its counter-receptors CD28 and CTLA-4. CD80 can be induced by the endotoxin lipopolysaccharide (LPS) via toll-like receptor four activation. LPS injection leads to transient podocyte foot-process effacement and proteinuria in mice, which is independent of lymphocytes because it also occurs in severe combined immune deficiency mice, which are devoid of lymphocytes. However, the LPS model is self-limiting as compared with the prolonged course of MCD in humans (*Reiser et al., 2004*; *Kistler & Reiser, 2010*). The mechanisms that promote persistent CD80 expression in MCD remain unknown. *Reiser & Mundel (2004)* speculated that CD80 induction in podocytes may be a physiological response to infection and facilitate the excretion of pathogens by transiently increasing the glomerular permeability to macromolecules. Modifying genetic or environmental factors may lead to persistent CD80 induction after a triggering event in MCD. *Garin et al. (2009)* postulated that MCD may be due in part to a defect in the ability of the immune system to turn off podocyte CD80 expression. Previous studies have suggested that soluble CTLA-4 produced by regulatory T cells can bind to dendritic cells expressing CD80 and act to block T-cell activation (*Taylor et al., 2004*). *Garin et al. (2009)* postulated that ineffective censoring of CD80 expression by T-regulatory cells may underlie the pathogenesis of MCD in light of the lower serum and urinary level of CTLA-4 in MCD patients in relapse. However, we found higher CTLA-4 levels in the serum of MCD patients in relapse versus remission, although these differences were not significant. In our study, CTLA-4 was absent in renal biopsies of patients in relapse, but present in those with partial remission. CTLA-4, which can turn off podocyte CD80 expression, may arise from renal tissue instead of blood circulation.

Both urinary CD80 and CTLA-4 levels differed between patients with steroid-sensitive and -resistant MCD in relapse. The urinary CD80 level was lower in patients with steroid-resistant than -sensitive MCD, but did not significantly differ from the urinary

CD80 levels in patients with IMN in relapse. When urinary CD80 levels are significantly higher and urinary CTLA-4 levels are lower, glucocorticoids therapy may achieve CR.

CD80 was still present in the glomeruli of patients with steroid-resistant MCD. Serum CD80 levels did not differ between patients with steroid-sensitive and -resistant MCD. However, CTLA-4 was present in both glomeruli and urine.

In comparing patients in relapse with steroid-resistant MCD and IMN, we found no significant differences in serum or urinary CD80 or CTLA-4 levels in renal tissues. Some IMN cases were steroid-resistant. CTLA-4 may fail to turn off some podocyte CD80 expression and therefore result in complete CD80 excretion to urine even with full-dose glucocorticoid therapy.

Early changes in gene expression could affect the course of primary glomerular disease (Clement et al., 2007). A decrease in expression of podocyte protein-tyrosine phosphatase (GLEPP1) was associated with partial steroid-sensitivity in several mouse models of podocyte disease, but, in contrast to GLEPP1, upregulated CD80 expression is steroid-resistant. Recent studies have found that CD80 is expressed in renal tissue in several types of glomerulonephritis, such as lupus nephritis, immunoglobulin A nephropathy, diabetic nephropathy, and Fabry disease (Reiser et al., 2004; Fiorina et al., 2014; Trimarchi et al., 2016; Wu et al., 2004; Sui et al., 2010), in addition to MCD. B7/CD28 blockade (LEA29Y, Belatacept) in kidney transplant recipients have proven that the replacement of toxic calcineurin inhibitor (CNI) use is feasible in selected populations. (D'Addio et al., 2013) Abatacept (CTLA-4–Ig fusion protein), a costimulatory inhibitor that targets CD80, induced partial or CR of proteinuria in patients with FSGS (Yu et al., 2013; Trimarchi, 2015), in which CD80 seemed to be minimally expressed in glomeruli and urine (Garin et al., 2010). However, few of these studies showed high urinary CD80 excretion in various nephritis diseases except MCD.

The method by which CD80 occurs in urine remains unknown. Previous studies have speculated that urinary CD80 presence may not reflect that CD80-positive podocytes are lost in the urine (Trimarchi et al., 2016; Petermann & Floege, 2007; Yu et al., 2005). CD80 may be contained in granular membrane structures found in urine during podocyte injury (Hara et al., 2005). Some studies have found that slit diaphragm proteins are shed into the urine (Gerke et al., 2005; Collino et al., 2008). CD80 that binds and sequesters slit diaphragm proteins may follow these proteins that are shed (Reiser et al., 2004). Why CD80 cannot be completely shed from podocytes in steroid-resistant MCD or other nephritis diseases, as in steroid-sensitive MCD, remains elusive. According to our data, dysfunction of CTLA-4 may play an important role in the pathogenesis of MCD, shedding light on further nephrotic research.

Nonetheless, we suggest that CD80 expression in renal tissue cannot be used to distinguish MCD from other nephritis diseases and cannot distinguish steroid-sensitive from -resistant MCD. However, CD80+/CTLA-4- expression on glomeruli may indicate the functional deficiency of T-regulatory cells (Cara-Fuentes et al., 2014a). Because of strongly positive expression in MCD patients and simultaneous negative CTLA-4 expression, glucocorticoid treatment might be effective. Increased urinary CD80 levels and reduced urinary CTLA-4 levels show a higher accessibility to remission

and better sensitivity of full-dose glucocorticoid therapy. Our study helps accelerate adult MCD therapy for at least 8 weeks by allowing doctors to prescribe immune depressive drugs together with glucocorticoids based on our proposed CD80 and CTLA-4 levels.

## CONCLUSIONS

In conclusion, for patients with MCD strongly positive CD80 expression and simultaneous negative CTLA-4 expression, or higher urinary CD80 level and lower urinary CTLA-4 level, glucocorticoids therapy may achieve CR. Urinary CD80 and CTLA-4 levels may play a role in diagnosis and prognosis as non-invasive biomarkers. Further studies investigating the precise mechanisms of the interaction of CD80 and CTLA-4 during the whole course of MCD are needed.

## ABBREVIATIONS

| | |
|---|---|
| **CTLA-4** | cytotoxic T-lymphocyte antigen-4 |
| **MCD** | minimal change disease |
| **INS** | idiopathic nephrotic syndrome |
| **IMN** | idiopathic membranous nephropathy |
| **FSGS** | focal segmental glomerulosclerosis |
| **ESRD** | end-stage renal disease |
| **CR** | complete remission |
| **uPCR** | urine protein:creatinine ratio |
| **LPS** | lipopolysaccharide |
| **SCID** | severe combined immune deficiency |
| **ROC** | receiver operating characteristic. |

## ACKNOWLEDGEMENTS

The authors thank all patients who participated in this study and the nurses from the nephrology department of Shandong Provincial Hospital affiliated to Shandong University.

### Funding

The authors received no funding for this work.

### Competing Interests

The authors declare that they have no competing interests.

### Author Contributions

- Bing Zhao performed the experiments, authored or reviewed drafts of the paper, approved the final draft.
- Hui Han contributed reagents/materials/analysis tools, prepared figures and/or tables, authored or reviewed drafts of the paper, approved the final draft.

- Junhui Zhen prepared figures and/or tables, authored or reviewed drafts of the paper, approved the final draft.
- Xiaowei Yang analyzed the data, prepared figures and/or tables, authored or reviewed drafts of the paper, approved the final draft.
- Jin Shang analyzed the data, authored or reviewed drafts of the paper, approved the final draft.
- Liang Xu contributed reagents/materials/analysis tools, prepared figures and/or tables, authored or reviewed drafts of the paper, approved the final draft.
- Rong Wang conceived and designed the experiments, authored or reviewed drafts of the paper, approved the final draft.

## Human Ethics

The following information was supplied relating to ethical approvals (i.e., approving body and any reference numbers):

The study was approved by the Institutional Review Board of Shandong Provincial Hospital affiliated to Shandong University (Ethical Application Ref No. 2014-022).

## Data Availability

The raw data are provided in the Supplemental File.

## Supplemental Information

Supplemental information for this article can be found online at http://dx.doi.org/10.7717/peerj.5400#supplemental-information.

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
