# Peer review of "CD80 and CTLA-4 as diagnostic and prognostic markers in adult-onset minimal change disease: a retrospective study"

_PeerJ, doi:10.7717/peerj.5400_

## Round 0.1 · original submission · Major Revisions

Thank you for submitting your manuscript to PeerJ. After careful consideration, we have decided that your manuscript must be substantially revised. Please do address all comments and issues raised thoroughly and unequivocally. Your revision must include a thorough discussion of reviewer #1's concerns.

·

Basic reporting

Written English used in the text is quite okay, but some sentences are unclear, i.e. page 3 line 5: "patients still return to full-dose ..." - what does this mean? do patients take steroids on their own?
Abstract: were detected CD80 ... does not make sense ... so please modify these essential parts and look for others.

I do have a problem with CD80 staining. We tried for 6 months to stain renal biopsies for CD80 and did not find any consistency (maximum 1 + staining). The recent Kidney Int paper by a French group highlights this issue (The costimulatory receptor B7-1 is not induced in injured podocytes). Thus, I am rather restricted towards further research on CD80, since the "main publications" are coming from similar working groups with clear connections.

Moreover, abatacept is not effective at all as has been shown by a recent JASN paper!

Data in the results part would benefit from sub-headings.

Experimental design

Primary research is well defined.

I would like to see confidence intervals which give us more impact than p-values. Moreover, multivariate analyses are missing, for example you should adapt your findings for baseline proteinuria.

Validity of the findings

The validity might be robust for people who believe in the important role of CD80.
I don't do so, thus I would not consider CD80 as a reliable marker. The presentation of data is accurate, but would benefit from sensitivity / specificity analyses (area under the curve). Moreover, confidence intervals should be given and adjustment for baseline factors to prove whether observed changes remain significant.

Additional comments

Dear Authors,

The paper is written in an acceptable English, however, changes should be made.

I have a few further comments:
1. p3l8: every patient should have renal biopsy, otherwise you can't diagnose your patient.
2. MCD prognosis is in almost 100% favorable in Europe, so I would not consider it as a malign disease. If FSGS is found in a second biopsy, the underlying histology is always FSGS.
3. phase I MN, what is this?
4. The course of steroids in your cohort is absolutely unclear. Which dosage, standardised tapering. 8 weeks is not a timeframe which is suggested by KDIGO, if you have other definitions, you need to state them accordingly. This is clearly missing.
5. Do you re-biopsy your patients once they relapse? This will likely show the same histology or am I wrong?

Reviewer 2 ·

Basic reporting

Please check general comments.

Experimental design

Please check general comments.

Validity of the findings

Please check general comments.

Additional comments

In this paper, 55 patients with biopsy-proven MCD and 26 patients with
biopsy-proven idiopathic membranous nephropathy (IMN), were detected
CD80 and cytotoxic T lymphocyte antigen-4 (CTLA-4) levels in serum,
urine and renal tissue. Authors claim that glucocorticoid treatment
may result in complete remission for only MCD patients with strongly
positive CD80 expression and negative CTLA-4 expression in glomeruli,
or higher urinary CD80 level and lower CTLA-4 level.

Major flaws:

- Authors stated that 55 patients with biopsy-proven MCD and 26
patients with biopsy-proven idiopathic membranous nephropathy (IMN)
were studied. Maybe it should be more appropriate to design a study
where groups are numerically balanced. Please explain this issue,
since it could lead to a statistical unbalance.

- In the Materials and Methods section, secondary antibodies employed
in Immunohistochemistry were not specified. Please fix this issue.

- Conclusions are too vague, please render them more accurate.

Minor flaws:

- In order to put your data in the context of literature, please
consider citing D’Addio et al. 2013 Plos One (CTLA-4 Ig), La Rocca et
al. 2000 Cell Transplant (kidney-pancreas tx) and Fiorina et al. 2001
Diabetes (kidney-pancreas tx in T1D).

---

## Round 0.2 · accepted · Accept

During the production process please implement the correction suggested by reviewer #1

# ·

Basic reporting

No comment.

Experimental design

No comment.

Validity of the findings

No comment.

Additional comments

Dear Authors,

Many thanks for the revision and replying to my queries in detail.

There is one reference which should be modified:
16 arin EH should be Garin EH ..

Best regards,
Andreas Kronbichler

Reviewer 2 ·

Basic reporting

After revision the paper has significantly improved in quality.

Experimental design

The description of methodology used is improved

Validity of the findings

the paper gives a significant contribution to MCD patients

Additional comments

The paper is now well written.